# Developing a Diagnostic Decision Support System for Benign Paroxysmal Positional Vertigo Using a Deep-Learning Model

**DOI:** 10.3390/jcm8050633

**Published:** 2019-05-08

**Authors:** Eun-Cheon Lim, Jeong Hye Park, Han Jae Jeon, Hyung-Jong Kim, Hyo-Jeong Lee, Chang-Geun Song, Sung Kwang Hong

**Affiliations:** 1Department of Otorhinolaryngology-Head and Neck Surgery, Hallym University College of Medicine, Anyang 14068, Korea; abysslover@gmail.com (E.-C.L.); pjh377@hanmail.net (J.H.P.); hjk1000@hallym.ac.kr (H.-J.K.); hyobravo@gmail.com (H.-J.L.); 2Laboratory of Brain & Cognitive Sciences for Convergence Medicine, Hallym University College of Medicine, Anyang 14068, Korea; 3Department of Convergence Software, Hallym University, Chuncheon 24252, Korea; wis7872@naver.com (H.J.J.); cgsong@hallym.ac.kr (C.-G.S.)

**Keywords:** vertigo, benign paroxysmal positional vertigo, deep learning, artificial intelligence

## Abstract

*Background*: Diagnosis of benign paroxysmal positional vertigo (BPPV) depends on the accurate interpretation of nystagmus induced by positional tests. However, difficulties in interpreting eye-movement often can arise in primary care practice or emergency room. We hypothesized that the use of machine learning would be helpful for the interpretation. *Methods*: From our clinical data warehouse, 91,778 nystagmus videos from 3467 patients with dizziness were obtained, in which the three-dimensional movement of nystagmus was annotated by four otologic experts. From each labeled video, 30 features changed into 255 grid images fed into the input layer of the neural network for the training dataset. For the model validation, video dataset of 3566 horizontal, 2068 vertical, and 720 torsional movements from 1005 patients with BPPV were collected. *Results*: The model had a sensitivity and specificity of 0.910 ± 0.036 and 0.919 ± 0.032 for horizontal nystagmus; of 0.879 ± 0.029 and 0.894 ± 0.025 for vertical nystagmus; and of 0.783 ± 0.040 and 0.799 ± 0.038 for torsional nystagmus, respectively. The affected canal was predicted with a sensitivity of 0.806 ± 0.010 and a specificity of 0.971 ± 0.003. *Conclusions*: As our deep-learning model had high sensitivity and specificity for the classification of nystagmus and localization of affected canal in patients with BPPV, it may have wide clinical applicability.

## 1. Introduction

Benign paroxysmal positional vertigo (BPPV) is characterized by brief recurrent vertigo with positional nystagmus triggered by a change in head position with respect to gravity. Therefore, a diagnosis of BPPV is based largely on the clinical features and patterns of nystagmus provoked by specific positional maneuvers, such as those in the Dix–Hallpike test and the supine roll test [1,2,3,4]. 

Large population-based studies revealed that 15–20% of healthy adults have suffered from dizziness yearly and the complaints annually account for about 5.6 million clinic visit in the United States [5,6], wherein BPPV was estimated the most frequent vestibular disease with an overall lifetime prevalence of 2.4% [7]. We concerned the majority of patients with BPPV visit either an emergency room or a primary care practice before they are referred to a neurotology specialist, wherein 65% of these patients undergo unnecessary diagnostic procedures, such as radiographic examination, even though the examination of nystagmus is critical to diagnose BPPV [8]. In addition, inappropriate treatment due to delay of diagnosis lead to a higher risk of falling or reduced physical activity in daily life [2,9]. Thus, given the notable prevalence, inappropriate interventions in patients with BPPV could be the leading cause financial burden of heath care system.

In recent years, machine-learning algorithms have been used to develop innovative diagnostic systems for medical image analysis [10]. For instance, the Inception V3 model (Google LLC, Menlo Park, California, USA) uses a mixed neural network of sub-convolutional neural network (CNN) structures to predict severity in cases of diabetic retinopathy, performing as well as human experts [11]. Interestingly, promising results have also been obtained by using deep learning to annotate scenes or detect spatiotemporal differences within videos [12], suggesting that deep-learning algorithms may be able to classify nystagmus patterns with spatiotemporal characteristics. Therefore, we hypothesized the diagnostic support system using machine leaning would be helpful when interpreting eye movements with spatiotemporal characteristics. 

BPPV is highly prevalent and diagnosing the BPPV or classifying affected canal of BPPV can be challenging in primary care practices. Additionally, when considering detailed examinations of eye movements have greater sensitivity than neuroimaging for differentiating central lesions in patients with isolated vertigo [13], the diagnostic support system for detecting eye movements might have wider application in emergency units as well. 

In this study, using an extensive set of nystagmus cases, we trained our deep-learning algorithm to identify three-dimensional eye movements and confirm involved semicircular canal of BPPV, and then validated the diagnostic performance of the model using clinical cases.

## 2. Methods

### 2.1. Data Collection 

To develop our algorithm, we reviewed the medical records of 90,451 patients who underwent the vestibular function test using infrared video goggles between January 2006 and February 2017. From these records, we selected 91,778 video clips recorded positional nystagmus from 3467 patients who reported symptoms of vertigo; the clips, which were reviewed by four otology experts, were classified according to a pattern of three-dimensional eye movements (horizontal, vertical and torsional) induced by ten positional tests, including bow and lean, lying down, the supine roll, head hanging, and the Dix–Hallpike test. All video clips were deidentified in clinical data warehouse, and an exemption was obtained from the institutional review board (Figure 1).

### 2.2. Preprocessing and Axis Measurement on Eye Movement

Individual eye movements from each video clip were scored as a vector sum of three-dimensional rotations in the horizontal, vertical, and torsional directions. The center of the pupil was tracked with an algorithm based on the center of gravity [14]. The circle Hough Transform was applied to detect ellipse-shaped pupils and identify eye-blinking [15]. If a pupil was found, edge-detection and ellipse-fitting algorithms were used to localize the center of the pupil. The directions of the horizontal and vertical movements were determined by the sum of the transient velocity, as previously described [14,16,17]. The center of the pupil in the first frame of each video clip was used as the reference.

Torsional changes were measured by cross-correlating the gray-level distributions derived from two sequential video frames along an arc of the iris [14]. Each pixel in the iris striations was projected to polar coordinates, thus converting a rotational movement to a linear metric. The projected iris image from a previous frame was used as the reference (Figure 2f), and the current image was used as the template (Figure 2g). Details of the calculations of transient and overall velocities can be found in the Appendix A. The algorithm determines whether the patient is normal by comparing the overall velocities with threshold values. If a normal state is detected, the algorithm will instantly terminate the process.

### 2.3. Training Datasets for Prediction of the Affected Canal in BPPV Cases 

We represented hand-crafted features of amplitude and direction of nystagmus as a grid image, which were obtained from the previous image processing stage and are fed to a simple 2D-CNN. Although we have changed the initial convolutional and long short-term memory (LSTM)-based fusion model to a simpler 2D-convolutional form, the core model can be replaced with a fully connected network. The model was built by Keras package [18] in python and tuned by changing depths of convolutional layers (1–12), the number of base neurons (2–512), and the number of random grid images (1000–1,000,000). We chose an 80:20 split for the training data. All incidence of different types of nystagmus in the training and test datasets were set to 12.5%, meaning that the number of incidence is equal for eight BPPV types. For example, if we selected 10,000 for the number of random grid images, there would be 10,000 training images and 1250 validation images for each BPPV type. In total, there would be 80,000 and 10,000 validataion images for eight BPPV types (Figure 3).

A grid image consisted of thirty rectangles (grids) (The columns of the grid image denote the results of ten positional test for BPPV, and the rows represent three axes movement of nystagmus; 3-demensional vector of nystagmus × 10 positional tests), each of which represented the amplitude of nystagmus. The amplitudes were converted to pixel values ranging from 0 to 255 and overall nystagmus amplitude from the video-dataset was adjusted to <1000, then normalized to 255. Each image was filled by gray pixels (value 128) or values randomly sampled from the univariate Gaussian distribution specific to a BPPV type. Details of the deep learning model using grid images for diagnsosis of BPPV can be found in the Appendix A.

### 2.4. Diagnostic Algorithm for BPPV and Test Datasets

The diagnosis of BPPV was based on a typical vertigo presentation that was associated with nystagmus induced by excitation of the semicircular canals by positional tests: torsional and upbeat nystagmus was induced by the Dix–Hallpike maneuver, which excites the posterior semicircular canal (PSC); horizontal nystagmus (geo- and ageotropic) was induced by the supine roll test, which excites the horizontal semicircular canal (HSC); and torsional and downbeat nystagmus was induced by the Dix–Hallpike on one or both side, or head-hanging test, which excites the anterior semicircular canal (ASC). Model validation for detecting nystagmus patterns and affected canals was conducted using video clips consisting of 3566 horizontal, 2068 vertical, and 720 torsional movements from 1005 patients with BPPV labeled by the otologic experts based on diagnostic criteria in Appendix A. The datasets for validation were selected with similar proportions of PSC, HSC and ASC BPPV as much as possible from our clinical data warehouse to avoid biased evaluation.

### 2.5. Statistical Analyses

Reference standards were set by the otologic experts. To evaluate model performance, we employed a one-vs.-rest multi-class strategy, in which the model regards a sample in one class as positive and all the others as negative. Confusion matrices were used to assess the overall accuracy of the model with the following formulae:
Accuracy = (TP + TN)/(TP + FP + TN + FN)Precision = TP/(TP + FP)Sensitivity = TP/(TP + FN)Specificity = TN/(TN + FP)False positive rate (FPR) = 1 − SpecificityF1 score = $\frac{\left(1+\beta^{2}\right)\times Precision\times Sensitivity}{\beta^{2}\times Precision+Sensitivity}$where TP = true positive, TN = true negative, FP = false positive, FN = false negative, and β = 1.


The micro-average of each metric was calculated by summing up individual TPs, FPs, TNs, and FNs of the different sets of predictions. By contrast, the macro-average was the mean of individual metrics. The non-information rate (NIR) is the rate of the most frequent class in the test dataset. Thus, NIR was used to determine the baseline accuracy of the model, as classifiers predicting the major class would have high accuracy if some datasets had a class imbalance. The one-tailed binomial test was used to determine if model accuracy were greater than the NIR. The area under the receiver operating characteristic curve (AUROC) was calculated to measure the impact of changes in cutoff values on prediction probability and model performance. Values were cross-validated and displayed as mean ± 95% confidence interval.

## 3. Results

### 3.1. Model Performance in Detecting Nystagmus Types

Model validation for detecting nystagmus patterns was conducted with video clips consisting of 3566 horizontal, 2068 vertical, and 720 torsional movements from 1005 patients with BPPV. Model AUROCs for horizontal, vertical, and torsional movements were 0.966, 0.952, and 0.853, respectively (Figure 4). When overall torsion was calculated using me an sample distance, the AUROC was 0.807 due to outlier effects. The sensitivity values were 0.910 ± 0.036 and 0.879 ± 0.029 for horizontal and vertical movements, respectively. The sensitivity of torsion detection was the lowest, at 0.783 ± 0.040. This low sensitivity may have resulted from the low pixel resolution of iris patterns, similar patterns in other iris areas, and errors that had accumulated during template matching. The specificity values for horizontal, vertical, and torsional movements were 0.919 ± 0.032, 0.894 ± 0.025, and 0.799 ± 0.038, respectively.

### 3.2. Model Performance in Identifying the Canal Affected by BPPV

Our trained model was used to predict the semicircular canal affected by BPPV in each case in the validation dataset. Model performance is summarized in Table 1. In comparison to the data annotations, our model had a macro-average AUROC of 0.901 ± 0.008 and a micro-average AUROC of 0.901 ± 0.007, indicating that model performance was excellent (Figure 5). The accuracy of the deep-learning model was 0.800 ± 0.008, which is significantly higher than the NIR of 0.168 (*p* value < 0.001).

## 4. Discussion

The diagnosis of acute vertigo is thought to be reliant on clinical features and analysis of eye movements. Detailed examinations of eye movements have also been shown to have greater sensitivity than neuroimaging for the purpose of differentiating early phase central lesions in patients with isolated vertigo [13]. Thus, we set the present work in the context of these initiatives. To do this, we focused on classifying eye movements using a deep-learning model and collated video clips of nystagmus from patients with BPPV, as it is the most common vestibular disease and the different types of nystagmus—horizontal, vertical, and torsional—are clearly observed.

Additionally, our approach could provide a substantial progress in public health care system because patients with BPPV frequently suffer from impaired quality of life due to delays in diagnosis or treatment [19] and the delays may occur when BPPV patients visit primary care practice or emergency departments even though the clinical treatment guidelines for BPPV are well-established [2,4,20,21]. In particular, only 10–20% of patients eventually receive canalith repositioning therapy [7,22], which suggest an automated diagnostic support system that classifies nystagmus types based on our initial idea would also be beneficial for primary care practice.

Model validation produced a sensitivity of 0.910 ± 0.036 and a specificity of 0.919 ± 0.032 for horizontal movement and a sensitivity of 0.879 ± 0.029 and a specificity of 0.894 ± 0.025 for vertical movement, suggesting acceptable model performance for nystagmus screening. However, detection of torsional movement by the model had lower sensitivity and specificity, at 0.783 ± 0.040 and 0.799 ± 0.038, respectively. This result may have been due to the low pixel resolution of iris patterns, similar patterns in other iris areas, and errors that accumulated during template matching. Sensitivity and specificity may be improved if the model were trained and validated using data from high-resolution video goggles. 

Methods for recording nystagmus using videonystagmography have been proposed previously [23,24,25,26]. However, our study goes beyond recording nystagmus and directly provides a method for classifying nystagmus types using a deep-learning architecture. 

We have initially tried to train a deep learning model of mixed convolutional and LSTM blocks, which are common in modelling action recognition tasks. The problem was that any spatio-temporal features of nystagmus had not been captured by different depth and combination of hidden layers. As a note, the fusion model was not better than random guesses in detecting the horizontal and vertical direction of nystagmus, thereby it could not determine the torsional counterpart as well. In turn, we decided to change the fusion model to a simpler form with a hand-crafted feature detector.

In another aspects, our work is superior to current technology of videonystagmography in that our algorithm could be embedded in any device that can record eye movement. Considering that eye tracking programs have been developed as a standard platform for a new generation of smartphones or laptops, our works can unlock the potential of mobile health technology.

In terms of predicting the affected canal in BPPV patients, the macro-average AUROC was 0.901 ± 0.008, and the micro-average AUROC was also 0.901 ± 0.007. However, because the accuracy of detecting torsional movements strongly affects the accuracy of diagnoses of PSC- and ASC-BPPV, the model sensitivity values for identifying PSC- and ASC-BPPV were relatively low. Additionlly, the model sensitivity differed between the right and left PSC-BPPV, with right PSC-BPPV misdiagnosed as left PSC-BPPV in 31.83 ± 1.20 cases. This may have resulted from errors in direction detection. Nevertheless, because horizontal and vertical nystagmus were detected with high accuracy, the model performed well in diagnosing cases of LSC-BPPV.

### Limitation and Future Research

We note that vestibular diseases other than BPPV were not included in the current model. Furthermore, our model cannot detect BPPV associated with multiple canals due to the nature of our datasets. Nevertheless, we expect that, after training with typical BPPV features, our model will be able to detect BPPV with involvement of multiple canal and atypical positional nystagmus that is not associated with BPPV. Therefore, as with automated electrocardiogram interpretation, our model could provide diagnostic support in primary care practices or emergency departments. In addition, even though our findings were resulted from our single institute, it is expected that the model can achieve much higher accuracy because our model was trained by random sampling of whole population of all possible nystagmus patterns may induce by positional test.

Future model refinements would improve the accuracy of the torsional detection by capturing torsional features with a recurrent neural network. Refining the time-series analysis would also help with identifying multiple canals affected in BPPV patients. The final end product would be a diagnostic support system based on deep-learning architecture that would be available in primary care practices for diagnosing patients with acute vestibular syndrome including BPPV.

## 5. Conclusions

Our deep-learning model had high sensitivity and specificity for identifying the affected canal in BPPV patients and for detecting nystagmus types. Further research will focus on validation in real clinical settings after model refinement. We hope that, if our deep-learning model can accurately diagnose positional nystagmus associated with BPPV, positional nystagmus that is not associated with BPPV could also be detected by the model. This would increase the utility of employing an automated diagnostic support system to examine patients with positional vertigo in primary care practices or emergency rooms.

## 6. Patents

Domestic patent (1018984140000), PCT/KR2018/004844, USA patent in progress.

## Figures and Tables

**Figure 1 jcm-08-00633-f001:**
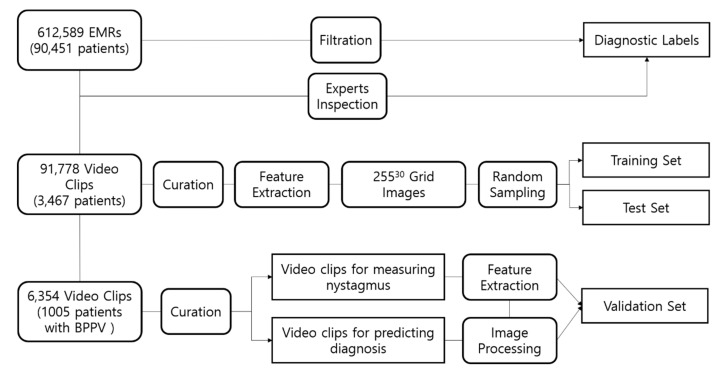
Data collection flow. Electronic medical records (EMRs) of 90,451 patients were filtered by keywords “vestibular function tests” and the video-clips recorded nystagmus were inspected by experts to obtain diagnostic labels for each patient. Among video-clips, those recorded nystagmus with typical movement on three axes were selected by experts and were used to create nystagmus directions for feature extraction (labels). Video clips of BPPV patients were used to generate the validation set for deep learning model. Grid images were randomly sampled using feature extraction, and the new set was split into training set and test set before training the model.

**Figure 2 jcm-08-00633-f002:**
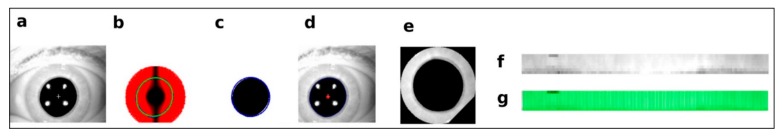
Image processing order. (**a**) Original frame, (**b**) circle Hough Transform followed by edge detection, (**c**) binarization by a threshold value, (**d**) pupil center and pupil area determined by ellipse fitting, (**e**) iris area selected, (**f**) iris image from the previous frame (reference image) after polar transformation (rotated 90° counterclockwise), and (**g**) iris image from the current frame (template image) with overlapping patches (rotated 90° counterclockwise).

**Figure 3 jcm-08-00633-f003:**
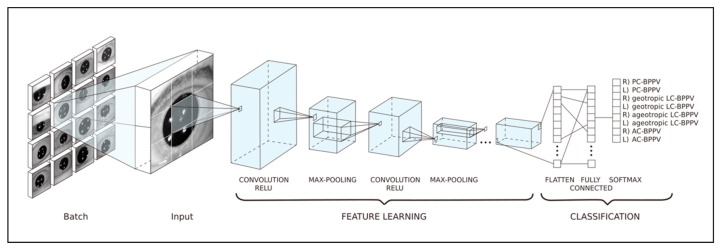
The convolution neural network architecture used to build the model for diagnosing benign paroxysmal positional vertigo (BPPV). From each video, 30 features were extracted using an algorithm to measure eye movement along three axes. These features were used as model inputs. In the model, a convolution layer, a batch-normalization layer, and a rectified linear unit activation layer were sequentially connected. The fully connected layer then employed the softmax function to diagnose BPPV types. (R = right, L = left, PC = Posterior semicircular canal, LC = Lateral semicircular canal, AC = Anterior semicircular canal).

**Figure 4 jcm-08-00633-f004:**
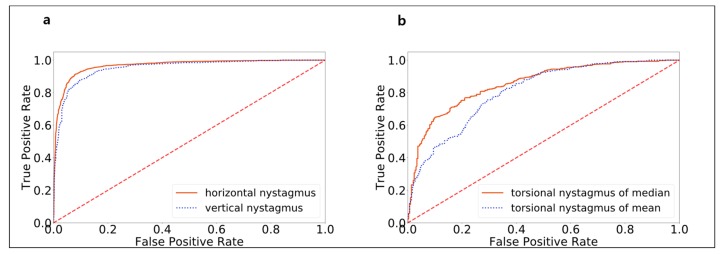
The receiver operating characteristic curve (ROC) of model performance in classifying nystagmus types after model training. The areas under (**a**) the ROCs for measuring horizontal and vertical nystagmus were 0.966 and 0.952, respectively, and the area under (**b**) the ROC for measuring torsional nystagmus was 0.853. When overall torsion was calculated using mean sample distance, the area under of the ROC was 0.807 due to outlier effects.

**Figure 5 jcm-08-00633-f005:**
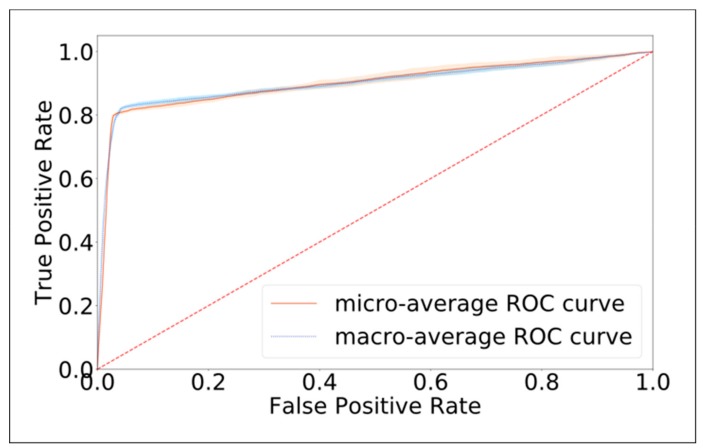
The receiver operating characteristic curve (ROC) of model performance in predicting the canal affected by benign paroxysmal positional vertigo (BPPV) after model training. The macro-average area under the ROC (AUROC) was 0.901 ± 0.008, and the micro-average AUROC was 0.901 ± 0.007, suggesting that our model performed well in identifying which canals were affected in BPPV patients.

**Table 1 jcm-08-00633-t001:** Model sensitivity and specificity in diagnosing types of benign paroxysmal positional vertigo (BPPV) after 10-fold cross-validation using nystagmus videos of 1005 patients.

Affected Canal	Precision	Sensitivity	Specificity	F1-score	N
R) PSC-BPPV	0.888 ± 0.014	0.560 ± 0.013	0.985 ± 0.002	0.686 ± 0.009	179
L) PSC-BPPV	0.753 ± 0.007	0.882 ± 0.005	0.952 ± 0.002	0.813 ± 0.005	144
R) geotropic LSC-BPPV	0.837 ± 0.009	0.916 ± 0.007	0.970 ± 0.002	0.875 ± 0.007	146
L) geotropic LSC-BPPV	0.771 ± 0017	0.941 ± 0.008	0.982 ± 0.002	0.847 ± 0.010	61
R) ageotropic LSC-BPPV	0.800 ± 0.012	0.957 ± 0.006	0.967 ± 0.003	0.871 ± 0.007	122
L) ageotropic LSC-BPPV	0.853 ± 0.010	0.934 ± 0.003	0.973 ± 0.002	0.891 ± 0.006	142
R) ASC-BPPV	0.775 ± 0.031	0.614 ± 0.015	0.986 ± 0.003	0.685 ± 0.015	73
L) ASC-BPPV	0.707 ± 0.022	0.663 ± 0.014	0.956 ± 0.005	0.684 ± 0.008	138
Average/Total	0.798 ± 0.015	0.808 ± 0.009	0.971 ± 0.002	0.794 ± 0.008	1005

R: right, L: left, PSC: posterior semicircular canal, LSC: lateral semicircular canal, ASC: anterior semicircular canal, N: number of samples.

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
