# Peer review of "Developing a Diagnostic Decision Support System for Benign Paroxysmal Positional Vertigo Using a Deep-Learning Model"

_jcm, 2019, doi:10.3390/jcm8050633_

Round 1

Reviewer 1 Report

Like this study, for the purpose of the investigation of the diagnositic accuracy, the gold standard for the diagnosis is very important. For the diagnosis of BPPV, the gold standard is the observation of the positional nystagmus. In the method part (page 4), they wrote "torsional and downbeat nystagmus was induced by the reverse Dix-Hallpike of head-hanging test, ----". For me, it is very strange. I have never read such definition of anterior canal type of BPPV. It is very common that torsional and downbeat nystagmus was induced by the reverse Dix-Hallpike test in patients with posterior canal type of BPPV, not anterior canal type of BPPV. I think they made wrong diagnosis to patients when they diagnosed anterior canal type of BPPV. The evidence in support of my idea is that the ratio of number of patients with anterior canal type of BPPV was too much high in this study (21%, 211/1005). Usually, the ratio of anterior canal type of BPPV aganst all BPPV was 1 - 5%. I have never seen the report that the ratio was more than 20%.  

Reviewer 2 Report

The authors have submitted a fascinating study in which they applied feature extraction algorithms to videomystagmography in patients with benign positional vertigo. These features were then fed into convolutional neural networks to automate detection of horizontal, vertical and torsional nystagmus. The authors presented impressive algorithm performance in identifying nystagmus and predicting the involved semicircular canal. I think this manuscript is certainly appropriate for publication, but would like to clarify a few points with the authors.

1. Please provide a table outline the incidence of different types of nystagmus in the training and test sets. Since all of the patients included in this study had some form of BPPV, there is the possibility that the CNN was biased toward predicting vertigo. For example, if the incidence of horizontal nystagmus was >90% among patients included in this set, this would bias the algorithm toward predicting horizontal nystagmus and in most cases this would be correct.

2. Please elaborate on how this algorithm could be implemented clinically. One of the limitations of this study that is not explicitly discussed is that no normal patients were included. Therefore this could not be used to detect nystagmus. Also, as the authors discussed, other types of nystagmus that are not caused by BPPV were not included. It seems to be that in its current form, it would be difficult for this algorithm to be implemented in an under-resourced area because 1) nystagmus has to be recognized and 2) a clinical diagnosis of BPPV must be made in order to identify patients in whom this automated diagnostic tool could be applied.

3. I presume since the authors used Keras packages that deep learning was performed using python, however this should be explicitly stated.

Reviewer 3 Report

Dear authors,

it was a great pleasure to read youd paper. It is well-written and important paper, both from the scientific and clinical point of view.

Overall, it's fine. I have just two minor suggestions:

Lines 56-57: sentence "BPPV is highly prevalent and they can be challenging condition seen in the primary care 56 practice" needs to be corrected.

Patients may be better described (clinical diagnosis and it's verification). As I understand, all patients were diagnosed BPPV, however, in the section "2. Methods"  it could be clarified.

Round 2

Reviewer 1 Report

I can understand your meaning of "reverse Dix-Hallpike test". And I can understand why the ratio of anterior canal type of BPPV was high. But the authors have to show the reason why the ratio of anterior canal type of BPPV was high in discussion part.

Author Response

Reviewer 1

I can understand your meaning of "reverse Dix-Hallpike test". And I can understand why the ratio of anterior canal type of BPPV was high. But the authors have to show the reason why the ratio of anterior canal type of BPPV was high in discussion part.

Reply: We thank the reviewer for this comment. We added the sentence according to your recommendation in the methods section.

 The datasets for validation were selected with similar proportions of PSC, HSC and ASC BPPV as much as possible from our clinical data warehouse to avoid biased evaluation. (Lines 145-147)